# Descriptive Acute Respiratory Distress Syndrome (ARDS) in adults with imported severe *Plasmodium falciparum* malaria: A 10 year-study in a Portuguese tertiary care hospital

**Luísa Graça**[1,2], **Isabel Gomes Abreu**[1,2], **Ana Sofia Santos**[1,2]*, **Luís Graça**[3,4], **Paulo Figueiredo Dias**[1,2], **Maria Lurdes Santos**[1,2]

1 Infectious Diseases Department, Centro Hospitalar Universitário São João, Porto, Portugal, 2 Faculdade de Medicina da Universidade do Porto, Porto, Portugal, 3 Escola Superior de Saúde do Instituto Politécnico de Viana do Castelo, Viana do Castelo, Portugal, 4 Unidade de Investigação em Ciências da Saúde: Enfermagem da Escola Superior de Enfermagem de Coimbra, Coimbra, Portugal

* asfaustino@gmail.com

**Data Availability Statement:** All relevant data are within the paper.

## Abstract

### Background

Acute respiratory distress syndrome (ARDS) is a severe complication of malaria that remains largely unstudied. We aim to describe the development of ARDS associated with severe P. falciparum malaria, its management and impact on clinical outcome.

### Methods

Retrospective observational study of adult patients admitted with severe P. falciparum malaria in an Intensive Care Unit (ICU) of a tertiary care hospital from Portugal from 2008 to 2018. A multivariate logistic regression analysis was used to identify factors associated with the development of ARDS, defined according to Berlin Criteria. Prognosis was assessed by case-fatality ratio, nosocomial infection and length of stay.

### Results

98 patients were enrolled, of which 32 (33%) developed ARDS, a median of 2 days after starting antimalarial medication (IQR 0–4, range 0–6). Length of stay in ICU and in hospital were significantly longer in patients who developed ARDS: 13 days (IQR 10–18) vs 3 days (IQR 2–5) and 21 days (IQR 15–30.5) vs 7 days (IQR 6–10), respectively. Overall case-fatality ratio in ICU was 4.1% and did not differ between groups. The risk of ARDS development is difficult to establish.

**Funding:** This research received no specific grant from any funding agency in the public, commercial, or not-for-profit sectors.

**Competing interests:** The authors have declared that no competing interests exist.

## Conclusion

ARDS is a hard to predict late complication of severe malaria. A low threshold for ICU admission and monitoring should be used. Ideally patients should be managed in a centre with experience and access to advanced techniques.

## Introduction

Malaria is still a frequent and potentially fatal disease that affects millions of people worldwide [1]. Although public health efforts are focused on reducing the incidence of infection and mortality in endemic countries, cases acquired through travelling that are diagnosed and treated in non-endemic countries also have its challenges. Due to its relatively low frequency the diagnosis of malaria may not be considered at first glance resulting in delayed diagnosis and treatment with an overall worse prognosis [2].

Severe cases of malaria are frequently caused by *Plasmodium falciparum*, although geographically dependent, and are associated with a plethora of complications, including acute respiratory distress syndrome (ARDS) [3]. Data on incidence, risk factors and outcome of ARDS in the context of malaria are largely unknown. This is related to a multitude of factors: there is limited access to intensive supportive care in places with high incidence of malaria (and its complications) [4]; non-endemic countries have less accumulated experience in dealing with severe malaria and the concept/definitions of ARDS vary largely between studies[5,6]. Nonetheless, ARDS has been reported in 5 to 25% of adult patients with severe *P. falciparum* malaria [7–9], with a directly related mortality ranging from 20% to 95% in developing countries [10–12].

The present study describes the development of ARDS associated with imported severe *P. falciparum* malaria its management and impact on clinical outcome.

## Methods

### Ethics statement

This study was approved by the institutional ethics committee (Comissão de Ética para a Saúde do Centro Hospitalar Universitário de São João, approval number 98/19) and was performed in accordance with the Declaration of Helsinki. Informed consent by participants was waived by the ethics committee.

### Study design and population

This is a retrospective observational study. We included all patients older than 18 years of age with a diagnosis of malaria due to *Plasmodium falciparum* who were treated in the Infectious Diseases Intensive Care Unit (ID-ICU) of Centro Hospitalar Universitário de São João, a tertiary teaching hospital from Porto, Portugal, between January 2008 and December 2018. Admissions to the ID-ICU for reasons unrelated to malaria diagnosis were excluded.

Patients were identified using the institutional electronic database and paper clinical records and data was collected retrospectively using a chart report form.

### Definitions and diagnosis criteria

Malaria diagnosis was made by immunochromatographic assay (Malaria Now-Binax®) and thin smear. Parasitaemia quantification was done whenever possible.

Patients were classified as having ARDS according to the Berlin Definition Criteria for ARDS [13]. Briefly, this classification implies that the following criteria are met: 1) ARDS occurs within 1 week of the onset of clinical symptoms or there are new or worsening respiratory symptoms; 2) Bilateral pulmonary opacities (visible on chest x-ray) are not completely explained by pleural effusion, lung collapse or nodules; 3) the cause of respiratory failure is not completely explained by cardiac failure or volume overload. Patients that meet these criteria can be further classified as having: mild ARDS, if partial pressure of oxygen in arterial blood/ fraction of inspired oxygen ratio (PaO2/FiO2) is $\leq$ 300 mmHg but > 200mmHg with a minimum positive end-expiratory pressure (PEEP) of 5 cm $H_2O$; moderate ARDS if PaO2/FiO2 was $\leq$ 200mmHg but > 100mmHg with a PEEP $\geq$ 5 cm $H_2O$ or severe ARDS if PaO2/FiO2 was $\leq$ 100mmHg with a PEEP of $\geq$ 5 cm $H_2O$.

Besides the 2015 World Health Organization (WHO) criteria for severe malaria cases [14], we assessed clinical severity on ICU admission using the Acute Physiology and Chronic Health Evaluation (APACHE) II score [15], Simplified Acute Physiology Score (SAPS) II [16] and the Sequential Organ Failure Assessment (SOFA) score [17].

Bacterial coinfections were defined based in clinical criteria and microbiologic cultures (blood cultures, urinary antigen) performed routinely at admission and whenever justified. According to epidemiology virus, such as zoonotic and influenza, were also investigated. Community-acquired co-infections were defined as any bacterial infection occurring within the first two days of hospitalization. Infections diagnosed later were classified as nosocomial.

Patient's outcome was evaluated as development of nosocomial infection during ICU stay, length of stay in ICU and total hospital time and all cause ICU and hospital mortality.

Subgroup analysis of these parameters between patients with and without ARDS was made.

## Statistical analysis

Continuous variables were compared among groups using the t test or Mann-Whitney test and categorical variables using the chi-squared test or Fisher's exact test. Pearson's correlation was used to test for variables related to the duration of mechanical ventilation. SPSS 25® was used for these evaluations. To identify independently associated variables with the development of ARDS we fitted a stepwise logistic regression model, using STATA 15.0 (StataCorp LP, Texas, USA).

## Results

### Patient demographics and clinical and laboratory characteristics

Between January 2008 and December 2018, 98 adult patients were admitted to the ID-ICU with severe malaria due to *P. falciparum*.

Table 1 shows the baseline characteristics of these patients. The two groups (ARDS and no ARDS) were similar concerning age, gender and comorbidities. The mean age was 43 ± 11 years old ranging from 18 to 68 years of age. Eighty-six patients (87.8%) were males.

Almost all patients were Portuguese (95; 96.9%) and 3 were from Angola (3.1%). About half the patients (43; 43.9%) were Portuguese emigrants working in Africa or on short businesses trips (47; 48%). The main country of acquisition was Angola (62; 63.2%) followed by Mozambique (15; 15%), Côte d'Ivoire (7; 7.1%), Democratic Republic of the Congo (3; 3.1%), Ghana, Guinea Bissau, Malawi (2 each; 2%) and Gabon, Benim, Ruanda, Senegal and Togo with 1 patient each (1%).

More than half the patients (59; 60.2%) had previous contact with *P. falciparum*: 43 (44.8%) had been living in an endemic area of malaria for at least two years at the time of the diagnosis

**Table 1. Demographic data for 98 patients with severe *Plasmodium falciparum* malaria admitted to the ID-ICU, with and without ARDS.**

| | No ARDS (n = 66) | ARDS (n = 32) | All (n = 98) | *P value* |
|---|---|---|---|---|
| **Demographics** | | | | |
| Age in years | 41.98 ± 10.61 | 45.30 ± 11.54 | 43.06 ± 10.97 | 0.161 |
| Male gender | 58/66 (87.9%) | 28/32 (87.5%) | 86/98 (87.8%) | 0.957 |
| **Reason for travel** | | | | |
| Immigration | 25/66 (37.9%) | 18/32 (56.3%) | 43/98 (43.9%) | |
| Business | 34/66 (51.5%) | 13/32 (40.6%) | 47/98 (48.0%) | |
| Tourism and VFR | 7/66 (10.6%) | 1/32 (3.1%) | 8/98 (8.2%) | |
| **Previous contact with malaria** | | | | |
| Previous diagnosis of malaria | 25/66 (37.9%) | 11/32 (34.4%) | 36/98 (36.7%) | 0.736 |
| Living in endemic area | 25/66 (37.9%) | 18/30 (60.6%) | 43/96 (44.8%) | 0.043 |
| **Comorbidities** | | | | |
| Respiratory disease † | 26/66 (39.4%) | 13/32 (40.6%) | 39/98 (39.8%) | 0.907 |
| Alcohol abuse | 6/66 (9.1%) | 1/32 (3.1%) | 7/98 (7.1%) | 0.282 |
| Immunosuppression‡ | 3/66 (4.5%) | 1/32 (3.1%) | 7/98 (7.1%) | 0.282 |
| Obesity | 4/66 (6.1%) | 1/32 (3.1%) | 5/98 (5.1%) | > 0.9995 |
| Diabetes | 4/66 (6.1%) | 1/32 (3.1%) | 5/98 (5.1%) | > 0.9995 |
| Pregnancy | 1/66 (1.5%) | 0/32 (0%) | 1/98 (1%) | > 0.9995 |
| Number of comorbidities | 1(0–1) | 0 (0–1) | 1 (0–1) | 0.252 |

Data are number (%), median (IQR) and mean ± SD

ARDS–acute respiratory distress syndrome; VFR–Visiting friends and relatives

†Thirty-nine (39.8%) patients had probable respiratory disease: 26 (26.5%) were current smokers, 10 (10.2%) were previous smokers, 2 (2%) had obstructive sleep apnea, 1 (1%) had Kartagener syndrome and 1 (1%) had asthma.

‡Four patients (4%) were immunocompromised: 2 (2%) had psoriasis; one (1%) had Still's disease under immunosuppressors, and 1 (1%) had chronic kidney disease under dialysis and previous history of cervical cancer.

and 36 (36.7%) had a former episode of malaria. Only four patients were taking chemoprophylaxis and all but one were taking it incorrectly.

Table 2 shows the clinical findings, laboratory characteristics and severity scores of these patients. Symptoms were reported for a median of 4.5 days (IQR 3–7), ranging from one to 15 days before diagnosis was made. Parasitaemia data was quantified in 93 patients (94.9%): the median parasitaemia was 8% (IQR 3–15), with a minimum of 1% and maximum of 50%. On admission, patients fulfilled a median of one (IQR 0–2, range 0–6) WHO criteria for severe *P. falciparum* malaria.

Arterial blood gases on admission were with a mean (±SD) PaO2 of 80.59 mmHg ± 15.87, range 46.4–120.4, and required a median FiO2 of 27% (IQR 21–45, range 21–100), resulting in a median ratio PaO2/FiO2 of 310.32 (IQR 174.53–369.05, range 83.7–429.05).

## Description of ARDS development

Thirty-two patients (33%) developed ARDS, all under antimalarial treatment, a median of 2 days after starting medication (IQR 0–4, range 0–6). Nineteen (59.4%) of these patients fulfilled ARDS criteria on admission to the ICU (mean time 1 day, IQR 0–3, range 0–5). At the moment of ARDS diagnosis, 24 patients (75%) still had a positive smear.

Clinical and analytical variables associated with the development of ARDS were: patients living in an endemic area (OR 2.46; CI 95% 1.02–5.95); respiratory rate >30 bpm (OR 2.87; CI 95% 1.00–8.24); albumin <3g/dL (OR4; CI95% 1.51–10.58); C-reactive protein >100mg/L (OR 3.63; CI95% 0.98–13.39); platelet count <50x10$^9$/L (OR 2.98; CI95% 1.13–7.84); need for

**Table 2. Clinical symptoms, laboratory parameters and severity scores for 98 patients with severe *Plasmodium falciparum* malaria, with and without ARDS.**

| | No ARDS (n = 66) | ARDS (n = 32) | All (n = 98) | *p value* |
|---|---|---|---|---|
| **Clinical symptoms** | | | | |
| Days of symptoms | 5 (3–7) | 4 (3–6) | 4.5 (3–7) | 0.595 |
| Cough | 14/66 (21.2%) | 12/32 (37.5%) | 26/98 (26.5%) | 0.087 |
| Respiratory rate >30 bpm | 6/53 (11.3%) | 8/25 (32%) | 14/78 (17.9%) | 0.026 |
| SatO2 < 96% | 9/55 (16.4%) | 8/25 (32%) | 17/80 (21.3%) | 0.113 |
| SAP < 90 mmHg | 3/62 (4.8%) | 7/29 (24.1%) | 10/91 (11%) | 0.006 |
| **Arterial blood gas test at admission** | | | | |
| PaCO2 < 35 mmHg | 51/59 (86.4%) | 25/32 (78.1%) | 76/91 (83.5%) | 0.976 |
| PaO2 < 60 mmHg | 4/59 (6.8%) | 2/32 (6.2%) | 6/91 (6.6%) | > 0.9995 |
| FiO2 ≥ 35% | 6/59 (10.2%) | 9/32 (28.1%) | 15/91 (16.5%) | 0.014 |
| Ratio PaO2/FiO2 < 300 | 10/59 (16.9%) | 13/32 (40.6%) | 23/91 (25.2%) | 0.002 |
| Arterial pH < 7.35 | 13/57 (5.3%) | 2/32 (6.2%) | 15/89 (16.8%) | > 0.9995 |
| Lactate > 2 | 20/59 (33.9%) | 15/32 (46.8%) | 31/91 (34%) | 0.108 |
| **Laboratory parameters** | | | | |
| Parasitaemia | | | | 0.981 |
| <2% | 6/63 (9.5%) | 3/30 (10%) | 9/93 (9.7%) | |
| 2 to 4% | 13/63 (20.6%) | 6/30 (20%) | 19/93 (20.4%) | |
| 4 to 10% | 17/63 (27%) | 7/30 (23.3%) | 24/93 (25.8%) | |
| ≥10% | 27/63 (42.9%) | 14/30 (46.7%) | 41/93 (44.1%) | |
| Leukocytes > 11x10$^9$/L | 5/66 (7.6%) | 4/32 (12.5%) | 9/98 (9.2%) | 0.429 |
| Albumin < 3 g/dL | 30/65 (46.2%) | 24/31 (77.4%) | 54/96 (56.3%) | 0.004 |
| LDH > 300 U/L | 51/63 (81.0%) | 27/29 (93.1%) | 78/92 (84.8%) | 0.132 |
| C-RP > 100 mg/L | 48/66 (72.7%) | 29/32 (90.6%) | 77/98 (78.6%) | 0.043 |
| Platelets (x 10$^9$/L) | 45.5 (25–80.5) | 34 (23–47.5) | 40 (25–69.25) | 0.002 |
| **Severity scores** | | | | |
| WHO criteria | | | | |
| Impaired consciousness | 5/66 (7.6%) | 4/32 (12.5%) | 9/98 (9.2%) | 0.429 |
| Multiple convulsions | 0/66 (0%) | 0/32 (0%) | 0/98 (0%) | |
| Acidosis | 13/57 (5.3%) | 2/32 (6.2%) | 15/89 (16.8%) | > 0.9995 |
| Hypoglycaemia | 0/66 (0%) | 1/32 (3.1%) | 1/98 (1%) | 0.149 |
| Severe anaemia | 3/66 (4.5%) | 0/32 (0%) | 3/98 (3.1%) | 0.549 |
| Renal impairment | 5/66 (7.6%) | 4/32 (12.5%) | 9/98 (9.2%) | 0.429 |
| Jaundice | 29/66 (43.9%) | 15/32 (46.9%) | 44/98 (44.9%) | 0.784 |
| Pulmonary oedema† | 6/53 (11.3%) | 8/25 (32%) | 14/78 (17.9%) | 0.026 |
| Significant bleeding | 4/66 (6.1%) | 1/32 (3.1%) | 5/98 (5.1%) | 0.536 |
| Shock | 0/66 (0%) | 1/32 (3.1%) | 1/98 (1%) | 0.327 |
| Hyperparasitaemia | 27/63 (42.9%) | 14/30 (46.7%) | 41/93 (44.1%) | 0.729 |
| Number of WHO criteria | 1 (0–2) | 1 (0.25–2) | 1 (0–2) | 0.212 |
| SAPS II | 20.5 (16–31.5) | 37.5 (26–53.5) | 26 (18–40) | < 0.0005 |
| APACHE II | 10 (7–15) | 15 (11–23) | 13 (8–17) | 0.001 |
| SOFA | 7 (5–10) | 12 (8–17) | 8 (6–12) | <0.0005 |

Data are number (%), median (IQR) and mean ± SD

ARDS–acute respiratory distress syndrome; bpm–breath per minute; SatO2 –peripheral oxygen saturation; SAP–systolic arterial pressure; PaCO2 –partial pressure of carbon dioxide in arterial blood; PaO2 –partial pressure of oxygen in arterial blood; FiO2 –fraction of inspired oxygen; LDH–lactate dehydrogenase; C-RP–C- reactive protein; SAPS II–simplified acute physiology score II; APACHE II–acute physiology and chronic health evaluation II; SOFA–sequential organ failure assessment.

† Pulmonary oedema was defined radiologically or by bronchoalveolar lavage

**Table 3. Treatment of 98 patients with severe *Plasmodium falciparum* malaria admitted to the ID-ICU, with and without ARDS.**

|  | No ARDS (n = 66) | ARDS (n = 32) | All (n = 98) | *p value* |
|---|---|---|---|---|
| IV artesunate | 11/66 (16.7%) | 4/32 (12.5%) | 15/98 (15.3%) | 0.593 |
| Days to negative smear | 3 (2–4) | 3 (3–4) | 3 (3–4) | 0.378 |
| Empirical antibiotics <48h from admission | 19/66 (28.8%) | 14/32 (43.8%) | 33/98 (33.7%) | 0.142 |
| Days of antibiotic therapy | 7 (6.75–7) | 7 (7–7) | 7 (7–7) | 0.066 |
| Vasopressors† | 12/66 (18.2%) | 28/32 (87.5%) | 40/98 (40.8%) | < 0.0005 |
| CRRT | 6/66 (9.1%) | 9/32 (28.1%) | 15/98 (15.3%) | 0.014 |

Data are number (%), median (IQR) and mean ± SD

ARDS–acute respiratory distress syndrome; IV–intravenous; CRRT–continuous renal replacement therapy

† Vasopressors were started to maintain mean arterial blood pressure above 65 mmHg

supplemental oxygen with a FiO2≥35% (OR 3.97; CI95% 1.25–12.60) and systolic arterial pressure (SAP) <90 mmHg (OR 6.26; CI95% 1.48–26.37).

## Overall management of patients with severe malaria in the ICU

Regarding anti-malarial medication, 15 (15.3%) patients were treated with intravenous artesunate while the remaining were treated with quinine dihydrochloride plus doxycycline (73) or clindamycin (10). The treatment scheme and days until negative smear did not differ between the ARDS and non-ARDS groups (Table 3).

Empirical antibiotics for presumed community acquired infections were started in 33 (33.7%) patients but only one had microbiologic confirmation: an urinary tract infection due to E. coli in ARDS group. The median duration of antibiotics was 7 days in both groups.

Eight patients in the no-ARDS group required mechanical ventilation for other reasons than respiratory failure. They all presented multiorgan disfunction with neurologic impairment.

## Management of respiratory failure

Of the 32 patients with ARDS, three (9.3%) required solely non-invasive ventilation (NIV) for a mean duration of 3 days.

Ten (32.3%) patients that were initially treated with NIV required intubation after a mean of 1.5 days (IQR 1–2.25) on NIV and needed a mean duration of IMV of 9.5 days (IQR 5.75–15.25).

In 19 (59.4%) patients the ventilatory support was only with invasive ventilation. Ten patients were intubated on ICU admission and the other 9 between day 1 and day 5. The median duration of support was 10 days (IQR 6–14).

Thirteen (40.6%) patients needed curarization. Three patient (9.4%) required prone positioning and 4 (12.5%) extracorporeal membrane oxygenation (ECMO) for 6, 7, 15 and 45 days.

An initial period of NIV was not associated with longer IMV (p = 0.890), higher PEEP (p = 0.697), higher FiO2 (p = 0.294) nor need for curarization (p = 0.705), prone positioning (p = 0.267) or ECMO (p = 0.592).

Another eight patients included in the no-ARDS group required invasive mechanical ventilation for other reasons rather than respiratory failure.

## Outcome

Table 4 shows the outcome of patients with and without ARDS. The median time of ICU stay was of 4.5 days (IQR 2–11.5, range 1–53) and the median length of hospitalization was 9 days

**Table 4. Outcome for 98 patients with severe *Plasmodium falciparum* malaria admitted to the ID-ICU, with and without ARDS.**

| | No ARDS (n = 66) | ARDS (n = 32) | All (n = 98) | *p value* |
|---|---|---|---|---|
| **Length of stay** | | | | |
| Days in ICU | 3 (2–5) | 13 (10–18) | 4.5 (2–11.25) | < 0.0005 |
| Days in hospital | 7 (6–10) | 21 (15–30.50) | 9 (7–18.5) | < 0.0005 |
| **Nosocomial infection** | | | | |
| Empirical antibiotics | 7/66 (10.6%) | 17/32 (53.1%) | 24/98 (24.5%) | < 0.0005 |
| Agent identified | 2/66 (3%) | 8/32 (25%) | 10/98 (10.2%) | 0.001 |
| Days of antibiotic | 7 (7–7) | 7 (7–10.75) | 7 (7–10) | 0.844 |
| VAP | | | | |
| Empirical antibiotics | 4/66 (6.1%) | 11/32 (34.4%) | 15/98 (15.3%) | < 0.0005 |
| Pathogen identified | 0/66 (0%) | 6/32 (18.8%) | 6/98 (6.1%) | 0.001 |
| **Case-fatality ratio** | 2/66 (3%) | 2/32 (6.3%) | 4/98 (4.1%) | 0.595 |

Data are number (%) and median (IQR)

ARDS–acute respiratory distress syndrome; ICU–intensive care unit; VAP–ventilator associated pneumonia.

(IQR 7–18.5, range 1–69). Both were significantly longer in patients who developed ARDS: 13 days (IQR 10–18) vs 3 days (IQR 2–5) and 21 days (IQR 15–30.5) vs 7 days (IQR 6–10), respectively.

Nosocomial infection was more common among patients with ARDS as well as the use of empirical antibiotics (OR 9.55; CI95% 3.35–27.21), ventilator associated pneumonia (VAP) (OR 8.12; CI95% 2.33–28.45) and identification of the etiologic agent for all infections (OR 10.67; CI95% 2.11–53.84).

Four patients died while being treated in the ICU, which corresponds to an overall case-fatality ratio in the ICU of 4.1%. Of these patients, three (3.1%) died shortly (1 to 2 days) after admission due to cerebral malaria. The other patient died 55 days after admission due to septic shock secondary to VAP. There were no in-hospital deaths following ICU discharge.

## Discussion

This cohort describes ARDS in severe *P. falciparum* malaria. One third of our patients met the criteria for ARDS, which is in accordance with the overall reported frequency in other studies [7,8,18]. Ten patients were included in non-ARDS group although with a PaO2/FiO2 $\leq$ 300 mmHg due to other malaria associated pulmonary pathology such as oedema or infection.

The risk of ARDS development is difficult to establish. The most cited score for ARDS prediction is the LIPS score (Lung Injury Prediction Score) [19,20]. However, this score does not consider malaria as a possible predisposing factor.

Predisposing risk factors previously described for ARDS, such as immunosuppression or pregnancy [21–23], did not arise in our cohort. However, this is most likely due to those characteristics being underrepresented among the travellers who caught malaria.

Although our experience is about ARDS in *Plasmodium falciparum* malaria it is known that, though rare, *P. vivax* malaria can also cause severe disease and *P. knowlesi* can cause severe forms in a particular geographic distribution with rare connection with our country.

Living in an endemic area or a previous episode of malaria didn't protect from ARDS in our cohort.

Clinical criteria also described as predictors of ARDS in LIPS score and in previous studies [24–26], such as a high respiratory rate (above 30 bpm), thrombocytopenia or hypoalbuminemia we didn't observe that association.

We could not find a correlation between duration of symptoms until initiation of antimalarial treatment or time until a negative smear and the development of ARDS. This may highlight the physiopathology of ARDS in malaria, where some of the pulmonary damage occurs as a consequence of the inflammatory cytokines and persistent inflammatory response of the host to infection, rather than a direct effect of high parasitaemia [27]. This can also explain why the use of intravenous artesunate did not have an apparent impact on the development of ARDS. Studies that led to the approval of intravenous artesunate in severe malaria also did not find such benefit [28].

The ability to define early clinical predictors of ARDS may be limited by our sample size. However, our experience suggests that patients with cough, respiratory rate > 30 bpm and in need of supplemental oxygen should be admitted in ICU and be monitored closely as a way of detecting early deterioration and development of ARDS.

In our cohort ARDS presented later in the course of the disease: it was diagnosed in a median of 2 days after starting antimalarial medication and severe ARDS occurred in a median of 2 days later than moderate ARDS. This most likely reflects the natural evolution of ARDS in malaria, which seems to be a late event in the course of the disease [7].

Patients with severe ARDS were intubated later which we believe reflects the longer time for the development of severe ARDS rather than missing the early signs of respiratory failure that led to more severe disease.

Even though patients scored low on WHO criteria, had a life threatening infection, evident by the high mortality risk scores at admission (estimated mortality of 25%, 37,5% and 92,5% for APACHE 2, SAPS II and SOFA, respectively) and presence of multiple organ dysfunctions besides ARDS. A threshold of 10% as parasitaemia criterion for severe malaria excludes many of the patients described above. Lowering the limit to 2% would identify 90.3% of our cohort as having severe malaria. Hence, in line with WHO treatment guidelines [14], we consider 2% parasitaemia as a criterion for complicated malaria in non-endemic settings and closely monitor all patients that fulfill this.

Regarding the management of respiratory failure, a non-invasive trial seems possible and can prevent unnecessary mechanical ventilation (and consequent complications) in these patients as long as it doesn't delay invasive ventilation.

Three patients were treated solely with NIV. The remaining ten patients required IMV after a NIV and did not have a worse outcome than those that were immediately intubated. Previous studies in severe malaria cases in patients with early ARDS development after antimalarial treatment also had positive results with a NIV trial [29].

Conversely, four patients with ARDS were put on ECMO due to severe refractory hypoxemia with a favourable outcome. Although the data on ECMO use on severe malaria is scarce, most case reports show a positive impact on the clinical evolution of these patients [30,31].

Patients with ARDS had a longer ICU and hospital stay and were more likely to have organ dysfunctions, such as renal or cardiovascular, and nosocomial infections including ventilator-associated pneumonia, which could contributed to the prolonged hospitalization.

Overall, ARDS mortality is described around 40 to 50% [6,32] and malaria associated ARDS mortality ranged from 20 to 95% in developing countries [10–12]. In our study, in-hospital case-fatality ratio was 6.3% in the ARDS group. Given that the primary cause for respiratory failure influences the prognosis of ARDS [19,20] it is possible that ARDS related malaria carries a better outcome. Nevertheless, young age and low comorbidity of our patients, timely recognition of respiratory distress, availability of an ICU inside the Infectious Diseases department and access to advanced techniques of invasive support probably justify the differences found between our results and what is overall described. Mortality did not differ between groups but the low number of deaths undermines our capacity to discriminate the impact of

ARDS on outcome. The importance of new ventilatory approaches in ARDS such as protective ventilation and, more recently, access to ECMO, probably explaining the reduced mortality comparing to our previous study [33].

In conclusion, ARDS in the context of malaria is a poorly understood subject, with several caveats regarding diagnosis, useful clinical predictors and overall optimal management. It is wise to closely monitor patients with subtle signs of respiratory distress, hypoxemia and/or a parasitaemia of at least 2%. The availability of a specialized Infectious Diseases ICU and current approaches to ARDS, including ECMO, influences the prognosis and clinical outcome of these patients and contributed to our low (4.1%) case fatality ratio.

The weakness of our study needs to be acknowledged. This is a retrospective study with a limited number of patients which could contribute to some unconclusive findings. Even so, it seems important to describe our experience in this subset of patients.

## Author Contributions

**Conceptualization:** Maria Lurdes Santos.

**Data curation:** Luísa Graça, Isabel Gomes Abreu.

**Formal analysis:** Luís Graça.

**Supervision:** Maria Lurdes Santos.

**Visualization:** Paulo Figueiredo Dias.

**Writing – original draft:** Luísa Graça, Isabel Gomes Abreu.

**Writing – review & editing:** Ana Sofia Santos, Maria Lurdes Santos.

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
