## [Decision Letter · Decision Letter 0]

27 Feb 2020

PONE-D-20-02362

Predictors of Acute Respiratory Distress Syndrome (ARDS) in adults with imported severe Plasmodium falciparum malaria: a 10 years study in a Portuguese tertiary care hospital

PLOS ONE

Dear Dr Santos,

Thank you for submitting your manuscript to PLoS ONE. After careful consideration, we felt that your manuscript requires revision, following which it can possibly be reconsidered. Although your manuscript was of interest to the three reviewers, major concerns were related to study design and data interpretation.  A methodological concern raised by the reviewers was related to   sample size as it may have compromised the conclusion.    Also, the authors should revise some distorted definitions; for example, the definition of semi-immunity.   Given the study limitations, it has been suggested that the purpose of the study should be adjusted from a “predictive” to a “descriptive” of ARDS among patients with severe imported malaria. Therefore, it seems to be a consensus that the word "predictors" should be removed from the title. For your guidance, a copy of the   reviewers' comments was included below

We would appreciate receiving your revised manuscript by March 30. To enhance the reproducibility of your results, we recommend that if applicable you deposit your laboratory protocols in protocols.io, where a protocol can be assigned its own identifier (DOI) such that it can be cited independently in the future. For instructions see: http://journals.plos.org/plosone/s/submission-guidelines#loc-laboratory-protocols

We look forward to receiving your revised manuscript.

Kind regards,

Luzia Helena Carvalho, Ph.D.

Academic Editor

PLOS ONE

Journal Requirements:

2. Please include your tables as part of your main manuscript and remove the individual files. Please note that supplementary tables (should remain/ be uploaded) as separate "supporting information" files

3. Please amend your manuscript to include your abstract after the title page.

4. Please upload a copy of Figure 1, to which you refer in your text on page 5. If the figure is no longer to be included as part of the submission please remove all reference to it within the text.

Reviewers' comments:

Reviewer's Responses to Questions

**Comments to the Author**

1. Is the manuscript technically sound, and do the data support the conclusions?

Reviewer #1: Yes

Reviewer #2: Partly

Reviewer #3: Partly

2. Has the statistical analysis been performed appropriately and rigorously? 

Reviewer #1: Yes

Reviewer #2: No

Reviewer #3: Yes

3. Have the authors made all data underlying the findings in their manuscript fully available?

Reviewer #1: No

Reviewer #2: No

Reviewer #3: Yes

4. Is the manuscript presented in an intelligible fashion and written in standard English?

Reviewer #1: Yes

Reviewer #2: No

Reviewer #3: Yes

5. Review Comments to the Author

Reviewer #1: The manuscript entitled “Predictors of Acute Respiratory Distress Syndrome (ARDS) in adults with imported severe Plasmodium falciparum malaria: a 10 years study in a Portuguese tertiary care hospital” presents an interesting set of data from malaria-associated ARDS patients, which contributes a great deal and are, in fact, very scarce in literature in this area. The work shows clinical, laboratory, hypoxemia and parasitemia data, comorbidities, treatments, among others. On the other hand, no substantial early predictive marker of ARDS has been identified, because PaO2/ FiO2 ratio <300 mmHg (already well established and known for ARDS) and SAP <90 mmHg, as the authors themselves reported in their conclusion “the absence of early predictors”. Therefore, I believe the word "predictors" should be removed from the title, as being overinterpretation, suggests a false idea of research presented to the reader.

I suggest MINOR REVISION.

Main points:

1- Flow diagram of cohort study detailing ARDS, no ARDS and exclusion criteria and also some data forest plot of the odds ratios should be used as it would be more interesting and attractive.

2- The authors should describe the results more clearly, following the format x% (y/z), as similar to data presented in the tables.

3- The authors could discuss a little about C-reactive protein, albumin, and platelets once these laboratory parameters as they appear differently between ARDS and non-ARDS patients.

4- Considering the fact that a greater number of ARDS-developing patients are semi-immune, it is interesting that the results be better discussed.

5- It is known the reported mortality in literature is very high concerning ARDS cases, but it is not shown in this manuscript. The authors could suggest (hypothesize) the reason why there was no difference in ARDS and non-ARDS cases, as well regarding SAP.

Minor points

1- “10 years study” must be written “10-year study”

2- What do “NIV” and “NIV trial” mean? The meaning of this acronym must appear when quoted for the first time, not in the end.

3- The acronym ECMO must be described in full when it appears first in the text.

4- What does UTI mean? Is it ICU?

5- Antibiotic treatment lasted 7 days in all groups. How do you explain p-value of 0.006 (table 3)?

6- “Nosocomial infections” and “Any infection” are shown in table 4. It is not clear if “any infection” means any nosocomial infection. Do the cases of community-acquired infections (<48h after admission) not appear in this table?

7- The tables should contain the type of statistical test used in each row or in legend.

Reviewer #2: Graca and coworkers examining retrospectively 98 patients with severe P. falciparum malaria admitted to their ID-ICU in Portugal between 2008-2018, aimed to identify “predictors” of ARDS development. However, their final conclusion is “ARDS is a hard to predict late complication of severe malaria”. I think that this study is flawed by two main issues: 1) the retrospective enrolment of patients; 2) the limited number of patients. Both contributed to their negative findings.

In their multivariate model only two variables were independently associated with development of ARDS: 1) PaO2/FIO2 < 300 mmHg); 2)SAP < 90 mmHg. However, the first one is now part of the ARDS Berlin definition (previous ALI , now mild ARDS) therefore should not be considered a risk factor for development of ARDS, and the second is, more properly, an indicator of severe malaria.

Given the above mentioned limitations I will suggest to change the purpose of the study from a “predictive” to a “descriptive” of ARDS among patients with severe imported malaria.

Results:

Subheadings

-(Patient demographic and clinical and laboratory characteristics):nationality of patients should be indicated and the area of acquisition of malaria specified. Former Portuguese colonies (either in results and table 1) is ambiguous and the reader is not required to know what they were.

- (Prediction of ARDS development): table 2 shows patients admitted to ID-ICU with (=32) and without ARDS (=66); however , as stated in the text “thirteen of these patients already fulfilled ARDS criteria on admission to the ICU”. The majority of pts developed ARDS during ICU stay (as reported in the literature and also in my experience) and therefore table should be changed accordingly.

It would be interesting to know parasitaemia value at the time of ARDS occurrence in comparison with other criteria (see the article by Marks ME et al. BMC Infect Dis 2013; 13:118).

Also in table 2 there is a discrepancy between “acidosis” (5/57, 0/28) and arterial pH < 7.35 (13/57, 2/29); a pH < 7,35 encompass the definition of acidosis (metabolic acidosis= plasma bicarbonate < 15 mmol /litre or pH < 7.35. WHO Guidelines for the treatment of malaria, 2nd Edition).

-(Management of respiratory failure): this is a descriptive paragraph difficult to follow; I suggest to try to summarize main messages in a table.

Discussion: It would be important to discuss the role of albumin level (high risk factor for both severe malaria and ARDS) (see article Bruneel F et al. Intensive Care Med 2016;42:1588-96)and malaria semi-immunity. Moreover, ARDS is (together with shock and acidosis) one of the factor associated with malaria death but in the present experience none of the patients died with/for ARDS. In a previous article from the same group (Malaria J 2012) regarding 59 pts with severe malaria admitted in the ICU from 2000-2011, 8/9 patients who died with malaria had ARDS. Can you comment on?

Although , your experience is about ARDS in P. falciparum malaria, it should be mentioned the role of this complication in P. vivax and P. knowlesi severe malaria.

Minor:

“Immunochromatogenic” change to ” immunochromatographic”

Reference 12 refers to India, actually to be considered an endemic country for malaria.

Table 1: seven patients were immunocompromised: 3 had HIV infection; I do not agree that HIV “per se” should be considered immunocompromission.

Reviewer #3: The paper by Graca et al is a retrospective study of ARDS in malaria patients, indicating that it is difficult to identify predictors for the development of this complication. It is very descriptive, the conclusions are not really novel, and some inaccuracies need to be corrected. Still the topic is important and it is useful to have a study on this complication from a well-equipped center.

Major comments:

1. The authors state that more ARDS is observed in semi-immune patients than in non-semi-immune. This is most likely wrong and based on a fairly misleading definition of semi-immunity in the methods section. What is meant by 'endemic area'? Semi-immunity only occurs in areas of high-exposure. Semi-immunity refers to protection from severe malaria, as a consequence of high exposure. I would suggest to use the terminology of 'semi-immunity' only if significant antimalarial immunity can be shown (e.g. antimalarial antibodies). If this cannot be proven, it is better to indicate that these patients were living for at least 2 years in an endemic area, without using the term semi-immunity. In fact, real semi-immunity might be supposed to protect from malaria ARDS, just like it does for other malaria complications.

2. It is surprising to note that several patients with PaO2/FiO2 < 300 mmHg are classified in the non-ARDS group. A better discussion of these patients is required. Do these patients correspond to those in the same group that have pulmonary edema (see table 2)? What was the cause of this pulmonary edema? Most likely these patients have also pulmonary pathology caused by malaria? It would be preferable to identify these patients as having malaria-associated pulmonary pathology, although they do not correspond perfectly to the Berlin definition of ARDS. I would suggest to discriminate these patients in a separate column in the tables and to discuss whether the Berlin criteria are or are not appropriate for malaria-related ARDS.

3. Result section: please indicate how many patients did develop ARDS before the start of antimalarial treatment.

4. Discussion: the first sentence is not correct. There are several studies in the literature available describing malaria patients developing ARDS. This has also been reviewed, e.g. Taylor et al. Chest 2012, Van den Steen et al. Trends Parasitol 2013, Mohan et al. J Vector Borne Dis 2008.

Minor comments:

Introduction: it is not true to state that severe malaria is almost universally caused by P. falciparum. This is geographically dependent; it is true in Africa but in other parts of the world P. vivax can also be a cause, and in some parts of South-East Asia (e.g. Borneo), P. knowlesi is the most important cause. The latter is particularly relevant for malaria-associated ARDS.

Introduction: it is wrong to state that ARDS has been reported in 5 to 25% of patients with severe Pf malaria. The great majority of severe malaria patients are children, who do not develop ARDS. This statement is only true for adult patients (and this study also includes only adults). This should be better stated.

Materials and methods: how were bacterial coinfections defined? Cultures of blood, bronchoalveolar lavages? Please define the methodology. Were viral co-infections (e.g. influenza, …) excluded or not? This should be mentioned.

Why are 87.8% of the patients males? Why this huge gender imbalance?

Some abbreviations should be defined, e.g. UTI, ECMO, EI

Result section: Eight patients in the no-ARDS required mechanical ventilation for other reasons than respiratory failure. Please define which other reasons.

Table 1: former Portuguese colonies in Africa: the countries should be named, not all biomedical scientists know all the details of Portuguese history.

Table 1. P value of 0.160 for reason to travel: what does this P value refers to? Comparison of which parameters?

Table 1: Previous diagnosis of malaria: indicate within which time frame

Table 1: smokers are classified as ‘respiratory disease’. While it is obvious that smoking contributes to respiratory disease, it might be better to differentiate the smokers from other respiratory diseases in a separate row in this table.

Table 2: How was pulmonary edema defined? Radiologically? Please define.

Table 3: use of vasopressors in the treatment: it would be preferable to indicate the number of patients with low blood pressure, or what were the thresholds or indications to start vasopressor therapy?

Define how volume overload was defined?

6. PLOS authors have the option to publish the peer review history of their article (what does this mean?). If published, this will include your full peer review and any attached files.

Reviewer #1: No

Reviewer #2: No

Reviewer #3: No

---

## [Author Response · Author response to Decision Letter 0]

7 Apr 2020

We wish to thank you for the review to our manuscript now entitled “Descriptive Acute Respiratory Distress Syndrome (ARDS) in adults with imported severe Plasmodium falciparum malaria: a 10 years study in a Portuguese tertiary care hospital”.

The authors agree with the points raised by the reviewers:

- The title and the purpose of the study was changed from "predictive" to "descriptive";

- The nationality of the patients was described as well as the area of acquisition of malaria;

- Data on Table 2 was corrected;

- We added the parasitemia value at the time of the ARDS occurence but we were not able to compare it with other criteria;

- The information described on the "Management of respiratory failure" paragraph was simplified for a better understanding;

- Discussion: we considered all the points suggested by the reviewers and changed the text accordantly; 

- We excluded HIV "per se" as a immunocompromission.

---

## [Decision Letter · Decision Letter 1]

12 May 2020

PONE-D-20-02362R1

Descriptive Acute Respiratory Distress Syndrome (ARDS) in adults with imported severe Plasmodium falciparum malaria: a 10 years study in a Portuguese tertiary care hospital

PLOS ONE

Dear  Dr. Santos,

Thank you for resubmitting your manuscript to PLoS ONE. Although the data from this study has potential to be informative, relevant topics raised by the reviewer #3 during the peer review process remain to be addressed by the authors. Unfortunately, the authors did not submit a rebuttal letter that responds to each point raised by the reviewers as required by the publication policy of PLoS journals. More Specifically, the authors should take into account relevant concerns such as  (i) comments on semi-immunity, including criteria to define immune status of their patients; (ii) the unreliable statement about the incidence of ARDS. At this time, we strongly suggest the authors to proper address all topics raised by the reviewers.  For your guidance, a copy of the reviewer’s comments was included below. 

We would appreciate receiving your revised manuscript by  June 10. To enhance the reproducibility of your results, we recommend that if applicable you deposit your laboratory protocols in protocols.io, where a protocol can be assigned its own identifier (DOI) such that it can be cited independently in the future. For instructions see: http://journals.plos.org/plosone/s/submission-guidelines#loc-laboratory-protocols

We look forward to receiving your revised manuscript.

Kind regards,

Luzia Helena Carvalho, Ph.D.

Academic Editor

PLOS ONE

Reviewers' comments:

Reviewer's Responses to Questions

**Comments to the Author**

1. If the authors have adequately addressed your comments raised in a previous round of review and you feel that this manuscript is now acceptable for publication, you may indicate that here to bypass the “Comments to the Author” section, enter your conflict of interest statement in the “Confidential to Editor” section, and submit your "Accept" recommendation.

Reviewer #1: All comments have been addressed

Reviewer #3: (No Response)

2. Is the manuscript technically sound, and do the data support the conclusions?

Reviewer #1: Yes

Reviewer #3: Partly

3. Has the statistical analysis been performed appropriately and rigorously? 

Reviewer #1: Yes

Reviewer #3: Yes

4. Have the authors made all data underlying the findings in their manuscript fully available?

Reviewer #1: No

Reviewer #3: Yes

5. Is the manuscript presented in an intelligible fashion and written in standard English?

Reviewer #1: Yes

Reviewer #3: Yes

6. Review Comments to the Author

Reviewer #1: The authors have adequately addressed most of the comments by the reviewers and I am satisfied with the quality of the paper in its present form.

Reviewer #3: Major comments:

1. The authors have to provide a decent point-by-point rebuttal to all of the referee comments, thereby copy-pasting each comment of each reviewer, with addition of their reply, including indication of the precise changes that were performed with page and line numbers. The referees are putting time and efforts in providing comments, the authors are requested to take the time and effort to provide a decent reply.

2. The authors have not taken our comment on semi-immunity seriously. Semi-immunity may easily take 5 years to develop, and is supposed to protect against severe malaria (=definition of semi-immunity). They should rephrase and remove this term from their manuscript. Also their new statement on partial immunity in the discussion is not correct. The authors did not provide any measurement on the antimalarial-immune status of these patients.

3. I don’t see any response to my second comment.

4. The erratic statement about the incidence of ARDS in malaria patients (my second minor comment) has not been corrected. See e.g. abstract of reference 7, ARDS is rare in children with malaria, and children are >90% of severe malaria patients. Children with severe malaria may develop respiratory distress due to completely different etiologies, e.g. acidosis, but this is not ARDS.

5. In addition to the problems indicated above, several of the other minor comments were also not addressed at all. This is chiefly unacceptable.

7. PLOS authors have the option to publish the peer review history of their article (what does this mean?). If published, this will include your full peer review and any attached files.

Reviewer #1: No

Reviewer #3: No

---

## [Author Response · Author response to Decision Letter 1]

29 May 2020

Ana Sofia Santos

Infectious Diseases Department, Hospital de São João, Porto, Portugal

asfaustino@gmail.com

29/05/2020

We wish to thank you for the review to our manuscript now entitled “Descriptive Acute Respiratory Distress Syndrome (ARDS) in adults with imported severe Plasmodium falciparum malaria: a 10 year-study in a Portuguese tertiary care hospital”. 

We would like to present our excuses because in the first review we haven’t respond to all points raised by the reviewers. In the mail sent with the revision we printed the word document attached and answered those questions but it was incomplete. We only realized that when we received the second mail.

We now believe we have addressed all the comments addressed.

The title and the purpose of the study was changed from "predictive" to "descriptive" considering that the paper wasn’t able to identify substantial early predictive marker of ARDS. 

The nationality of the patients was described as well as the area of acquisition of malaria. The concept of the Former Portuguese colonies was removed.

The paragraph “Management of respiratory failure” was simplified for a better understanding. 

The type of statistical test used is described in methods.

On Table 4 any infection was related to any nosocomial infection but it wasn’t clear. This line was removed.

P value on antibiotic treatment is 0.066 and not 0.006, and so, there is no difference between groups.

Table 2 data on patients with p/F < 300 on admission was corrected as well as patients with acidosis.

Even though we have patient albumin levels we weren’t able to relate it to malaria semi-immunity, as suggested by reviewer #2. 

In this review none on the patients died with ARDS in severe malaria and a question was raised because in a previous study of our group 8/9 patients died. We think the new ventilatory approaches in ARDS such as protective ventilation and access to ECMO may explain the reduced mortality. 

We make a brief mention on the role of ARDS in P. vivax and knowlesi malaria in our description.

HIV "per se" as a immunocompromission was excluded considering the count of T CD4+ lymphocytes was above 500 cells/mm3 in all three HIV patients. 

The definition of semi-immunity was removed from the manuscript as we couldn’t prove antimalarial immunity. As so, the conclusion that ARDS is more observed in semi-immune patients rather than in non-semi-immune patients was withdrawn because it could lead to misleading interpretations. Instead we considered a subgroup of patients living in endemic areas for at least 2 years. 

We included 10 patients with PaO2/fiO2 < 300 in the non-ARDS group because they didn’t meet all the criteria for ARDS namely the exclusion of volume overload or concomitant infection as cause of respiratory failure. 

All patients developed ARDS under antimalarial treatment.

We added the parasitemia value at the time of the ARDS occurence but we were not able to compare it with other criteria.

The first sentence of the Discussion was corrected because it was inaccurate. 

In Introduction we changed the sentence that stated that P. falciparum was responsible for almost every cases of severe malaria. Our experience is only with P. falciparum. The incidence of 5-25% of patients with severe P. falciparum malaria is in adult patients, the statement was corrected.

Bacterial coinfections were defined based in clinical criteria and microbiologic cultures performed routinely at admission and whenever justified. According to epidemiology, virus, such as zoonotic and influenza, were also investigated.

There was a gender imbalance (87.8% males) justified by the reason of travel: Portuguese emigrants working in Africa or on short businesses trips. 

The abbreviations were defined. 

Eight patients in the no-ARDS group required mechanical ventilation for other reasons than respiratory failure. They all presented multiorgan disfunction with neurologic impairment.

P value for reason to travel was removed on table 1. Under the table is the description of the smokers and other respiratory diseases.

On table 2 was added how pulmonary oedema was defined: radiologically or by bronchoalveolar lavage.

The threshold to start vasopressor therapy was mean arterial pressure of 65 mmHg. It is defined on the legend of Table 3.

Volume overload was defined based on clinical manifestations, echocardiography and radiologically. 

We didn’t have data available on the time frame of the previous diagnosis of malaria. 

We appreciate your time and look forward to your response. Please address all correspondence concerning this manuscript to me at asfaustino@gmail.com.

Sincerely,

Ana Sofia Santos

---

## [Decision Letter · Decision Letter 2]

16 Jun 2020

Descriptive Acute Respiratory Distress Syndrome (ARDS) in adults with imported severe Plasmodium falciparum malaria: a 10 year-study in a Portuguese tertiary care hospital

PONE-D-20-02362R2

Dear Dr.  Santos,

We’re pleased to inform you that your manuscript has been judged scientifically suitable for publication and will be formally accepted for publication once it meets all outstanding technical requirements.

Kind regards,

Luzia Helena Carvalho, Ph.D.

Academic Editor

PLOS ONE

Additional Editor Comments (optional):

Reviewers' comments:

Reviewer's Responses to Questions

**Comments to the Author**

1. If the authors have adequately addressed your comments raised in a previous round of review and you feel that this manuscript is now acceptable for publication, you may indicate that here to bypass the “Comments to the Author” section, enter your conflict of interest statement in the “Confidential to Editor” section, and submit your "Accept" recommendation.

Reviewer #1: All comments have been addressed

Reviewer #3: All comments have been addressed

2. Is the manuscript technically sound, and do the data support the conclusions?

Reviewer #1: Yes

Reviewer #3: Yes

3. Has the statistical analysis been performed appropriately and rigorously? 

Reviewer #1: I Don't Know

Reviewer #3: Yes

4. Have the authors made all data underlying the findings in their manuscript fully available?

Reviewer #1: No

Reviewer #3: Yes

5. Is the manuscript presented in an intelligible fashion and written in standard English?

Reviewer #1: Yes

Reviewer #3: Yes

6. Review Comments to the Author

Reviewer #1: The authors have adequately addressed most of the comments by the reviewers and I am satisfied with the quality of the paper in its present form.

Reviewer #3: (No Response)

7. PLOS authors have the option to publish the peer review history of their article (what does this mean?). If published, this will include your full peer review and any attached files.

Reviewer #1: No

Reviewer #3: No

---

## [Editor Report · Acceptance letter]

25 Jun 2020

PONE-D-20-02362R2 

Descriptive Acute Respiratory Distress Syndrome (ARDS) in adults with imported severe *Plasmodium falciparum* malaria: a 10 year-study in a Portuguese tertiary care hospital  

Dear Dr. Santos:

I'm pleased to inform you that your manuscript has been deemed suitable for publication in PLOS ONE. Congratulations! Your manuscript is now with our production department. 

Kind regards, 

on behalf of

Dr. Luzia Helena Carvalho 

Academic Editor

PLOS ONE